# Hematological Markers in Thromboembolic Events: A Comparative Study of COVID-19 and Non-COVID-19 Hospitalized Patients

**DOI:** 10.3390/jcm14093192

**Published:** 2025-05-05

**Authors:** Elena María Gázquez-Aguilera, Tesifón Parrón-Carreño, Delia Cristóbal-Cañadas, Bruno José Nievas-Soriano, David Lozano-Paniagua

**Affiliations:** 1Department of Internal Medicine, Torrecárdenas University Hospital, 04009 Almería, Spain; elena.gaz.agui@gmail.com; 2Department of Nursing, Physiotherapy, and Medicine, Faculty of Health Sciences, University of Almería, 04120 Almería, Spain; tpc468@ual.es (T.P.-C.); dlozano@ual.es (D.L.-P.); 3Neonatal and Paediatric Intensive Care Unit, Torrecárdenas University Hospital, 04009 Almería, Spain; dcc380@ual.es

**Keywords:** COVID-19, thrombosis, inflammation mediators, blood coagulation disorders

## Abstract

**Background/Objectives**: SARS-CoV-2 infection increases thrombotic events in hospitalized patients, especially those of greater severity. It has been associated with the cytokine storm and worsening renal and liver function, increased inflammatory markers, and altered coagulation markers. This study analyzes differences in inflammatory, hepatic, renal, and coagulation markers between hospitalized patients with and without COVID-19 who experienced thromboembolic events during the last three years of the pandemic. **Methods**: This single-center, retrospective observational study, with an inferential component and biomarker analysis, included 663 patients (600 without COVID-19, 63 with COVID-19) admitted between December 2022 and January 2023. **Results**: Patients with COVID-19 exhibited significantly higher mean glomerular filtration rate (GFR) (100.5 mL/min/1.73 m^2^; *p* < 0.01) and alanine aminotransferase (ALT) levels (33.0 IU/L; *p* < 0.01) compared to those without COVID-19. Ferritin levels were also significantly elevated in COVID-19 patients (441.1; *p* < 0.01), particularly those with severe disease. Conversely, troponin I was significantly higher in patients without COVID-19 (22.6 × 10^4^ pg/mL; *p* < 0.001). Among COVID-19 patients, D-dimer levels were significantly higher in those not requiring intensive care unit (ICU) admission (9.0 × 10^3^ ng/mL; *p* = 0.023). Multivariate analysis revealed a significant association between COVID-19 and sex. **Conclusions**: Overall, renal function did not differ significantly between COVID-19 and non-COVID-19 patients. However, renal function was better in patients admitted to the ICU, regardless of COVID-19 status. Troponin I levels were elevated in non-COVID-19 patients, while ferritin and ALT levels were higher in COVID-19 patients. D-dimer levels showed no significant difference between the two groups.

## 1. Introduction

In late 2019, Chinese authorities alerted the World Health Organization (WHO) to atypical pneumonia cases in Wuhan, Hubei province. By January 2020, the causative agent was identified as a new coronavirus, the severe acute respiratory syndrome type 2 coronavirus (SARS-CoV-2), leading to COVID-19. WHO declared it a pandemic in March 2020, sparking a global health crisis [1]. As of May 2023, the WHO no longer categorizes it as a pandemic due to widespread vaccination and self-testing but advises continued vigilance [2]. SARS-CoV-2 primarily impacts the respiratory system but also leads to a high incidence of thrombosis, affecting both arterial and venous systems. Critically ill patients, particularly those in intensive care units (ICUs), are especially prone to thrombotic complications [3].

A recent systematic review and meta-analysis found the prevalence of venous thromboembolism (VTE) in COVID-19 patients to be 7.9% in non-ICU admissions and 22.7% in ICU admissions, with corresponding pulmonary embolism prevalences of 3.5% and 13.7%, respectively [4]. Pulmonary embolism (PE) contributes to the majority of venous thromboembolism (VTE), followed by deep vein thrombosis (DVT) [5]. The International Society on Thrombosis and Haemostasis (ISTH) and The American Society of Hematology (ASH) have established guidelines to standardize reporting criteria for coagulation-related events and bleeding in COVID-19. These guidelines promote data sharing and comparison, enabling healthcare professionals to tailor treatments to disease severity and determine whether observed thrombosis results from viral infection or the inflammatory response [6].

Recent studies have highlighted a significant correlation between COVID-19 and vascular coagulopathy [7], potentially stemming from dysregulated molecular pathways during disease progression. Reports also suggest a link between coagulopathy and disseminated intravascular coagulation (DIC), particularly in severe COVID-19 cases. Retrospective cohort studies consistently show elevated D-dimer and fibrin/fibrinogen degradation products in COVID-19 patients, underscoring the pivotal role of fibrin formation in this coagulopathy [8].

COVID-19 can disrupt both innate and adaptive immunity. Severe cases are characterized by dysregulated innate immune responses and cytokine storms, contributing to disease pathogenesis. This dysregulation triggers immunothrombosis pathways, resulting in clot formation [9]. Along with the cytokine storm, ferritin and C-reactive protein (CRP) levels drive proinflammatory pathways by activating the NF-κB transcription factor. This activation enhances the expression of proinflammatory mediators, contributing to disease inflammation and coagulopathy development [10]. In this context, a feedback loop exists among ferritin, CRP, and cytokines, suggesting that targeting ferritin could potentially mitigate the onset of the proinflammatory cascade [11]. Modulating ferritin’s biological effects could influence key aspects of atherothrombosis and venous thromboembolism pathogenesis, such as platelet activation and complement system activation via the classical pathway [12].

Biochemically, patients with poorer COVID-19 prognosis demonstrate increased liver and renal dysfunction [13] and elevated levels of inflammatory markers (ferritin, CRP, IL-6), coagulation markers (D-dimer, fibrinogen), cardiac markers (troponin I, LDH), and platelets, alongside distinct lipidomic changes [14]. Despite anticoagulant prophylaxis, hospitalized patients remain at high risk of thromboembolic disease [15]. Although numerous studies have examined biomarkers associated with thromboembolic events in COVID-19 patients, few have directly compared them to non-COVID-19 cases. While existing predictive models assist in venous thromboembolic risk assessment, further independent evaluation of both venous and arterial events is necessary to strengthen the evidence base for these markers [16]. This study aimed to compare inflammation (IL-6, ferritin, troponin I, CRP, LDH), hepatic (AST, ALT, GGT), renal (creatinine, GFR), and coagulation (D-dimer, fibrinogen, platelets) biomarkers between two cohorts of hospitalized patients with and without COVID-19 who experienced thromboembolic events (venous and arterial) at a tertiary medical center during the pandemic.

## 2. Materials and Methods

### 2.1. Study Population and Data Collection

A retrospective, single-center descriptive observational study with an inferential component was conducted to analyze data from 663 patients who experienced a thromboembolic event during their admission to Torrecárdenas University Hospital (Almeria, Spain) between January 2020 and October 2022. Patients were identified through the Diraya platform, the electronic health database of the Andalusian Health System (SAS), using search terms such as thrombosis, thrombus, pulmonary thromboembolism (PTE), deep vein thrombosis (DVT), and superficial vein thrombosis (SVT). Data collection for this study took place between December 2022 and January 2023

This study included hospitalized patients aged 18 years or older who experienced a new thromboembolic event, regardless of SARS-CoV-2 infection status. Exclusion criteria were age under 18, in-hospital death, evaluation only in outpatient or emergency departments, prior thromboembolic events, and refusal to participate.

Patients were divided into two groups: those with COVID-19 and those with a thromboembolic event (COVID-19 patients) and those with a thromboembolic event but admitted for other reasons (non-COVID patients). The study was approved by the Research Ethics Committee of Torrecárdenas University Hospital (reference 128/2022, 30 November 2022) and was conducted following the Declaration of Helsinki. All personal data were processed according to applicable legal regulations.

### 2.2. Statistical Analysis

Statistical analysis was performed using SPSS version 28 (IBM Inc., Armonk, NY, USA) with a significance level of 0.05. Descriptive analysis included calculating arithmetic means and standard deviations for quantitative variables and frequencies and percentages for categorical variables. The Kolmogorov–Smirnov test was utilized to assess the distribution of quantitative variables, revealing a non-normal distribution. Consequently, non-parametric tests, specifically the Mann–Whitney U, Kruskal–Wallis, and Chi-square tests, were employed to determine the statistical significance of differences between the COVID-19 and non-COVID-19 groups. Spearman’s rank correlation analysis was conducted to evaluate linear associations between biomarkers within both study cohorts. Separate statistical analyses were performed for each subgroup to examine potential variations in biomarker associations related to the distinct pathophysiologies of arterial versus venous thromboembolic events. Finally, multiple binary logistic regression analysis, adjusted for covariates identified as significant in bivariate analysis, was used to evaluate the risk of ICU admission. The variance inflation factor (VIF) was calculated for each independent variable to preclude multicollinearity, with VIF > 10 indicating severe collinearity.

## 3. Results

The baseline characteristics of the study population, categorized by admission with thromboembolic events related to COVID-19 or other etiologies, are summarized in Table 1. A total of 663 patients were included, comprising 600 non-COVID-19 patients and 63 COVID-19 patients. No statistically significant differences were noted in age or sex distribution between the two cohorts. The patient population was predominantly Spanish. Specifically, 95.7% of the non-COVID-19 group did not exhibit severe COVID-19 criteria. ICU admission was necessary for 8.6% of the overall patient population, with a significantly higher proportion observed in the non-COVID-19 group. No significant differences in prior COVID-19 infection were identified between the COVID-19 and non-COVID-19 groups.

Table 2 presents the bivariate analysis results comparing biomarker levels between the COVID-19 and non-COVID-19 patient groups. Significantly higher mean GFR was observed in the COVID-19 group. Only ALT demonstrated a statistically significant difference among liver function markers, with higher mean values in the COVID-19 group. AST, GGT, and LDH levels did not differ significantly between the groups. Ferritin levels were significantly elevated in COVID-19 patients, whereas troponin I levels were significantly higher in non-COVID-19 patients. No statistically significant differences were found for the inflammatory markers CRP and IL-6 or the coagulation markers platelets, fibrinogen, and D-dimer.

The Spearman correlation analysis in both cohorts (Figure 1 and Figure 2) revealed a pattern of significant associations between various biochemical markers. Negative correlations of GFR with creatinine and troponin I and a positive association with ALT and platelet count were identified. Additionally, significant positive correlations were observed between liver enzymes (AST and ALT), markers of liver damage and cholestasis (AST with GGT, ALT with GGT), markers of cellular damage (AST and ALT with LDH), markers of iron deposition (AST and ALT with ferritin), and a cardiac marker (ALT with troponin I), although in this case, the association is negative. Furthermore, the inflammatory marker CRP showed a positive correlation with fibrinogen, and other relevant positive associations were identified between creatinine and troponin I, GGT with LDH and ferritin, platelets with fibrinogen, and LDH with ferritin.

The analysis of the non-COVID-19 cohort (Figure 1) showed several significant correlations: AST was positively associated with troponin I, CRP, IL-6, and D-Dimer; creatinine with IL-6 and platelets; and GFR with GGT, while GFR was negatively correlated with IL-6. Positive correlations were also observed between GGT and CRP; LDH and CRP and D-Dimer; ferritin and IL-6, CRP, and D-Dimer; troponin I and D-Dimer; and CRP with IL-6, platelets, and fibrinogen.

On the other hand, in the analysis of the COVID-19 patient cohort (Figure 2), significant positive correlations were observed between AST and platelets and between ALT and platelets. In contrast, significant negative correlations were found between ALT and CRP and between troponin I and platelets.

Appendix A detail biomarker levels categorized by sociodemographic factors (age and sex) and clinical characteristics (severe COVID-19 and ICU admission). Appendix A compares renal function (Creatinine, GFR) and liver function (ALT, AST, GGT) parameters between COVID-19 and non-COVID-19 patients. Age significantly affected creatinine and GFR levels (*p* < 0.001). While severe COVID-19 status showed a near-significant effect on creatinine and GFR (*p* = 0.094 and *p* = 0.069, respectively), ICU admission significantly impacted GFR, ALT, and AST levels (*p* = 0.010, *p* < 0.001, and *p* < 0.001, respectively). Sex did not significantly influence renal or liver function parameters. Appendix A compares inflammatory parameters (LDH, Ferritin, Troponin, CRP, IL-6) between COVID-19 and non-COVID-19 patients. Significant differences were observed in ferritin levels across age groups (*p* = 0.015) and troponin I levels across age groups (*p* < 0.001) and sex (*p* = 0.040). Severe COVID-19 showed higher ferritin and CRP but lower Troponin and IL-6 than non-severe cases (*p* = 0.004). ICU admission was associated with higher troponin I but lower IL-6 levels (*p* < 0.001). Appendix A illustrates the comparative analysis of coagulation parameters (Platelets, Fibrinogen, and D-dimer). Age significantly influenced fibrinogen levels (*p* = 0.047) and D-dimer levels (*p* = 0.002). Sex affected platelet counts (*p* = 0.035) and D-dimer levels (*p* = 0.028). Severe COVID-19 status did not show a significant impact on coagulation parameters. ICU admission significantly affected D-dimer levels (*p* = 0.023).

Table 3 and Table 4 present the biomarker differences between COVID-19 and non-COVID-19 patients hospitalized due to thromboembolic events, analyzed separately for each vascular territory, arterial and venous. Table 3 reveals that hospitalized COVID-19 patients with arterial events presented significantly lower creatinine and higher GFR but elevated ALT, ferritin, and troponin I compared to non-COVID-19 patients. Platelet counts, fibrinogen, and D-dimer were similar. These findings suggest potential renal and liver function alterations, alongside increased inflammation and cardiac injury in COVID-19 arterial events.

In contrast, analysis of venous events (Table 4) showed a near-significant trend for lower creatinine (*p* = 0.056) and significantly higher platelet counts in COVID-19 patients. At the same time, other biochemical and inflammatory markers were broadly comparable between the groups.

Multiple binary logistic regression analysis was performed to identify factors predictive of ICU admission, and the results are presented in Table 5. The models were adjusted for the following independent variables: sex, age, prior COVID-19 infection, creatinine, GFR, AST, ALT, GGT, LDH, ferritin, troponin I, CRP, IL-6, platelets, fibrinogen, and D-dimer. In the COVID-19 patient group, male sex was found to be a significant predictor of ICU admission, with an odds ratio (OR) of 2.131 (*p* = 0.023). The Nagelkerke R^2^ values, indicating the model’s goodness of fit, were 0.620 for COVID-19 patients and 0.388 for non-COVID-19 patients. To evaluate the presence of multicollinearity among the covariates included in the multivariable logistic regression model, the VIFs were calculated for each variable. The VIF values ranged from 1.1 to 6.2, considerably below the commonly accepted threshold of 10. These findings confirm the absence of significant multicollinearity, ensuring the reliability of the regression estimates.

## 4. Discussion

This study compared biomarkers related to venous and arterial thromboembolic events in hospitalized COVID-19 and non-COVID-19 patients. While prior studies predominantly examined venous thrombosis due to its increased frequency, this research included patients with events in both venous and arterial territories. Recognizing that systemic involvement in COVID-19 significantly increases morbidity and mortality, predictive scoring systems, such as COMPASS 19, are being developed to aid in the identification of patients at increased risk, utilizing platelet count, prothrombin time, antithrombin levels, CRP, D-dimer, total lymphocyte count, and hemoglobin [17]. In our cohort, we have selected creatinine, glomerular filtration rate (CKD-EPI formula), AST, ALT, GGT, LDH, ferritin, CRP, troponin I, IL-6, platelets, fibrinogen, and D-dimer as hematological markers.

It is well-established that acute kidney injury develops in 30–50% of hospitalized COVID-19 patients, with increased incidence in those requiring intensive care [18,19]. Contrary to this, we observed a higher GFR in COVID-19 patients admitted to the ICU than in non-ICU patients. This seemingly paradoxical finding may be explained by the older age and higher comorbidity burden in the ICU cohort, potentially masking or overriding the expected decline in GFR associated with acute kidney injury. Another set of factors that could explain the higher GFR in COVID-19 patients, especially in the ICU, could be attributed to clinical management and the pathophysiology of critical illness. Firstly, optimized anticoagulation, crucial for preventing thrombotic events like renal microthrombosis, is associated with better outcomes and potentially preserved renal function [20]. Similarly, appropriate hemodynamic management with judicious use of vasopressors and strict fluid balance monitoring can prevent renal hypoperfusion and overload, both detrimental to kidney function [21]. To this is added the implementation of specific therapies and advanced ventilatory support strategies like protective ventilation, prone positioning, and extracorporeal membrane oxygenation (ECMO), which can improve tissue oxygenation and reduce respiratory stress, indirectly protecting renal function [22]. Lastly, genetic factors and individual variability in inflammatory and hemodynamic responses to SARS-CoV-2 might also play a role in preserving renal function in this population [23,24].

In patients with chronic kidney disease (CKD) and COVID-19, activation of the receptor for advanced glycation end products (RAGE) may initiate or exacerbate renal injury, potentially leading to thrombotic complications [25]. Neutrophil extracellular traps (NETs) also contribute to coagulation-related complications, particularly in CKD patients [26]. A prospective study of 701 COVID-19 patients demonstrated that elevated baseline serum creatinine significantly increased the risk of acute kidney injury and death, possibly due to hematogenous dissemination and viral accumulation in the kidneys, causing renal cell necrosis [27]. Consistent with the known impact of age on renal function, our study found that the glomerular filtration rate (GFR) declined with age in both COVID-19 and non-COVID-19 patients.

Despite thromboembolic events, creatinine values and glomerular filtration rate remained within the normal range in COVID-19 and non-COVID-19 patients. However, a negative correlation was observed between renal function markers and troponin I and IL-6, which are associated with COVID-19 infection. These findings likely reflect sample heterogeneity and the influence of other risk factors for venous thromboembolism (VTE).

Patients with liver disease face increased risks of both bleeding and thrombosis. Hospitalized patients with cirrhosis exhibit a twofold increased risk of venous thromboembolism (VTE) compared to those without cirrhosis [28]. Studies have shown a correlation between coagulation and liver damage in COVID-19, with elevated AST and ALT associated with coagulopathy markers [29,30]. Consistent with these findings, our study observed correlations between elevated ALT and GGT, LDH, ferritin, and troponin in non-COVID-19 patients and with GGT, LDH, ferritin, CRP, and platelets in COVID-19 patients, supporting the link between liver damage, coagulopathy, and endotheliopathy, but not IL-6. Notably, AST and ferritin were highest in non-COVID-19 patients aged 40–64, while GGT was more elevated in younger patients.

Prior research has proposed a thromboinflammatory mechanism connecting liver injury in COVID-19 to endotheliopathy, potentially mediated by IL-6 trans-signaling and subsequent elevation of coagulation factors [31]. Our study observed a significant correlation between IL-6 and fibrinogen exclusively in non-COVID-19 patients, supporting this mechanism in that group. Although interleukin-6 (IL-6) is recognized for its role in the inflammatory cascade and its contribution to endothelial injury, coagulopathy, and thromboembolism [32], its blockade is considered a potential strategy to attenuate cytokine storms [33], we did not identify a statistically significant difference in mean IL-6 levels between the COVID-19 and non-COVID-19 groups, despite the observed correlation with fibrinogen.

Myocardial injury is a frequent complication among hospitalized individuals. COVID-19 infection may induce an imbalance between the heightened metabolic demands of the infection and diminished cardiac reserve, indicating the potential utility of serial troponin measurements as a prognostic indicator [34]. Compared to patients with severe pneumonia from other causes, those with SARS-CoV-2 infection exhibit increased thrombophilia and myocardial damage, as evidenced by elevated levels of various biomarkers [35]. However, in contrast to these findings, our study observed higher mean troponin levels in non-COVID-19 patients. Furthermore, the most significant associations with troponin elevation were identified as D-dimer in non-COVID-19 patients and platelet count in COVID-19 patients, suggesting potential variations in the underlying mechanisms of myocardial injury between the two groups.

Unlike other studies, this study found elevated troponin I levels in non-COVID-19 patients, which could be attributed to the higher prevalence of comorbidities and cardiovascular risk factors in this group. Chronic ischemic heart disease, smoking, anticoagulant use, and liver disease could induce or exacerbate troponin release [36]. Non-cardiac surgical interventions, prolonged immobilization, and sepsis, more frequent in non-COVID-19 patients, also contribute to myocardial damage and troponin elevation. In contrast, although COVID-19 can elevate troponin, associated comorbidities are less common in this group [37]. Therefore, the elevated troponin I in non-COVID-19 patients in this cohort appears to be more related to their cardiovascular risk profile and comorbidities than to the viral infection itself.

The intense inflammatory response to COVID-19 can induce a hypercoagulable state, potentially leading to disseminated intravascular coagulation (DIC) and multiple organ dysfunction [38]. Elevated D-dimer at admission is associated with critical illness, death, and thrombohemorrhagic complications [39]. In our study, COVID-19 patients without ICU admission and female patients exhibited higher D-dimer levels, though these differences were not statistically significant compared to non-COVID-19 patients. This may be due to sample size limitations and variations in vascular territories, as D-dimer increases were less pronounced than CRP increases. Furthermore, in non-COVID-19 patients, D-dimer increased with age over 65, possibly reflecting lower ICU admission rates and impaired D-dimer clearance due to reduced glomerular filtration in older patients.

While routine D-dimer monitoring lacks prospective validation, guidelines recommend its use within the clinical context for VTE screening [40]. Conversely, elevated serum CRP is considered a reliable indicator of SARS-CoV-2 infection severity and is associated with increased VTE risk [41]. This association may stem from CRP’s role in immunity, complement activation, and prothrombotic effects [42]. CRP-induced microvascular inflammation can lead to endothelial activation and a prothrombotic state [43,44]. Patients with CRP above the median are more susceptible to VTE and acute kidney injury. Although bivariate analysis showed no significant difference in mean CRP levels between COVID-19 and non-COVID-19 patients, CRP positively correlated with platelet count, fibrinogen, D-dimer, LDH, ferritin, IL-6, AST, and GGT, all markers of coagulopathy, thrombosis, and acute phase response.

The cytokine storm associated with COVID-19 triggers the release of ferritin from hepatocytes, Kupffer cells, and macrophages under the influence of inflammatory cytokines [45]. This dysregulated immune response, manifesting as macrophage activation and hyperferritinemia syndrome, can contribute to a thrombotic storm and subsequent multiple organ damage, thereby increasing the severity of COVID-19. In alignment with prior research, our study demonstrated elevated ferritin levels in patients meeting the criteria for severe COVID-19. Furthermore, increased ferritin levels were more frequently observed in male and non-COVID-19 patients.

A meta-analysis has shown that COVID-19 patients with thrombotic complications have significantly higher ferritin levels, linking thrombosis to a hyperinflammatory state [46]. Consistent with this, our study found elevated mean ferritin levels in COVID-19 patients. In non-COVID-19 patients, ferritin correlated with inflammatory markers (CRP, IL-6), fibrinogen, and liver enzymes (AST, ALT, GGT). Conversely, in COVID-19 patients, ferritin primarily correlated with LDH and liver enzymes (AST, ALT, GGT).

Platelets are central to inflammatory signaling and immune responses [41], playing a key role in hemostasis and infection defense [47]. A meta-analyses of COVID-19 patients [48] report that platelet counts can vary widely, with thrombocytopenia and thrombocytosis observed in 13–61% of severe cases. Other meta-analyses state that patients with severe COVID-19 have a significantly lower platelet count [49,50]. In contrast, our study found normal mean platelet counts in COVID-19 and non-COVID-19 patients. Notably, COVID-19 patients under 40 and male non-COVID-19 patients had higher platelet counts. These findings diverge from the typical association of thrombocytopenia with severe COVID-19 [51], suggesting demographic factors and disease severity may influence platelet counts.

In this line, a recent study by Qiu et al. utilized single-cell transcriptomics and machine learning to examine platelet subpopulations in COVID-19, sepsis, and systemic lupus erythematosus [52]. The authors identified transcriptional platelet alterations linked to an increased risk of endotheliopathy and coagulation in fatal cases and potential impacts on lymphocyte function. These findings underscore the pivotal role of platelets in the pathophysiology of severe inflammatory diseases and pave the way for new research opportunities, supporting the notion that platelet count, activation, and functionality may serve as significant clinical determinants.

Lactate contributes to tissue damage by activating metalloproteinases and promoting macrophage-mediated angiogenesis [53]. Elevated lactate dehydrogenase (LDH) levels are frequently observed in COVID-19 patients, particularly those with severe disease, and LDH is increasingly recognized as a clinically relevant biomarker for assessing COVID-19 severity [50,51,53,54]. In our analysis, we observed a higher mean LDH level in COVID-19 patients compared to non-COVID-19 patients. A notable increase in LDH levels was observed among African patients within the COVID-19 group, although this difference did not reach statistical significance.

While decreased fibrinogen, alongside increased prothrombin time and D-dimer, is associated with disseminated intravascular coagulation (DIC) in severe COVID-19 [55], paradoxically, very high fibrinogen is seen in less severe prothrombotic COVID-19 [56]. Our study did not observe a statistically significant difference in fibrinogen levels between the COVID-19 and non-COVID-19 patient groups. However, we did find that in COVID-19 patients, fibrinogen levels primarily correlated with CRP, whereas in non-COVID-19 patients, they demonstrated correlations with CRP, troponin I, IL-6, and platelet count.

Analysis of thrombotic territories in COVID-19 patients revealed notable differences in various biomarkers. In cases of arterial thrombosis, alterations in creatinine, GFR, ALT, ferritin, and troponin were identified, likely attributable to systemic inflammation, endothelial dysfunction, and coagulopathy induced by COVID-19. Conversely, venous thrombosis exhibited prominent elevations in creatinine and platelet counts, which may correlate with platelet hyperreactivity and acute kidney injury linked to SARS-CoV-2 infection. In contrast, thrombosis among non-COVID-19 patients tended to follow a more traditional multifactorial profile [21,57]. Interestingly, D-dimer levels did not show significant differences between the two groups in either territory. This observation may stem from distinct underlying mechanisms and the effects of comorbidities and treatments in the non-COVID-19 cohort [58]. These findings underscore that thrombosis associated with COVID-19 represents a unique clinical phenotype driven by specific immunothrombotic mechanisms.

A logistic regression analysis of hospitalized COVID-19 patients in New York demonstrated that, after adjusting for multiple variables, men exhibited a significantly greater probability of thrombosis and mortality compared to women (36% vs. 29%, *p* < 0.001) [59]. These results are consistent with our findings, which also reveal a significant association between male sex and an increased likelihood of adverse clinical outcomes.

This study possesses several notable strengths. Firstly, it evaluated biomarkers in venous and arterial territories, a methodological approach rarely employed in comparable research. Secondly, it is the first study to compare patients with and without SARS-CoV-2 infection biomarker profiles directly. Thirdly, the relatively large sample size contributes to the robustness and validity of the findings. However, the study also presents limitations. One notable limitation of this study is the disproportion in sample sizes between the non-COVID-19 group (*n* = 600) and the COVID-19 group (*n* = 63). This imbalance, stemming from the methodological and contextual challenges posed by the COVID-19 pandemic during the recruitment phase, as highlighted in the literature [60], may have restricted the statistical power available to identify subtle differences in specific biomarker analyses. Nonetheless, the significant findings achieved, despite this imbalance, offer valuable and reassuring insights. It is important to interpret these results cautiously, considering the potential implications of limited statistical power. Additionally, the lack of autopsy data for deceased patients limited the investigation of specific causes of death. Finally, the cross-sectional design precludes the establishment of causal relationships and does not include long-term patient follow-up.

## 5. Conclusions

Our study suggests renal function remains unaffected, mainly in COVID-19 and non-COVID-19 patients with thrombosis. However, specific analytical parameters, including ALT, GGT, troponin I, and IL-6 in non-COVID-19 patients, and ALT, troponin I, and platelets in COVID-19 patients, may demonstrate significant interrelationships. Notably, COVID-19 patients with thrombosis exhibited higher mean levels of the liver enzyme ALT, suggesting a potential association between liver damage, coagulopathy, and endotheliopathy. Elevated ferritin levels were observed in both COVID-19 and non-COVID-19 patients with thrombosis, and these levels appeared to correlate with inflammatory markers, such as CRP and IL-6, and coagulation markers, including fibrinogen. Furthermore, troponin I levels were elevated in non-COVID-19 patients with thrombosis and may be associated with coagulation markers, specifically D-dimer in non-COVID-19 patients and platelets in COVID-19 patients. Further research is warranted to fully elucidate the complex associations between these biochemical markers, COVID-19, and thrombosis. Integrating individualized risk assessment models and biomarker monitoring into thromboprophylaxis protocols can improve patient outcomes by tailoring prophylactic strategies to the specific risk profiles of hospitalized patients. A case-control study design could provide a more precise evaluation of these relationships, potentially leading to improved therapeutic protocols for managing these patient populations.

## Figures and Tables

**Figure 1 jcm-14-03192-f001:**
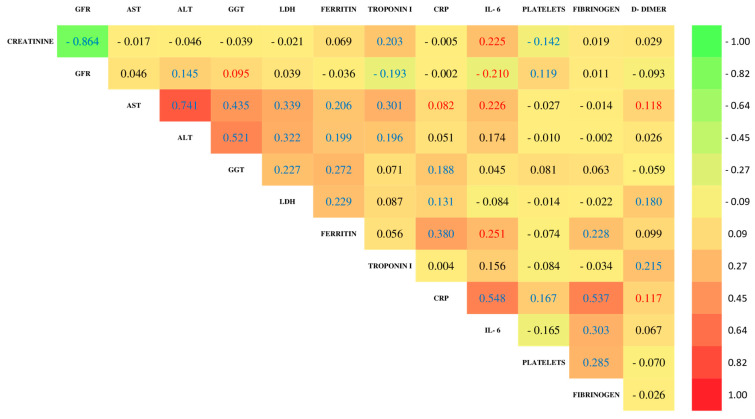
Spearman’s rank correlation coefficients for clinical variables in non-COVID-19 patients (level of significance: *p* < 0.01 are marked in blue, and *p* < 0.05 are in red. Non-significant values with *p* ≥ 0.05 are marked in black).

**Figure 2 jcm-14-03192-f002:**
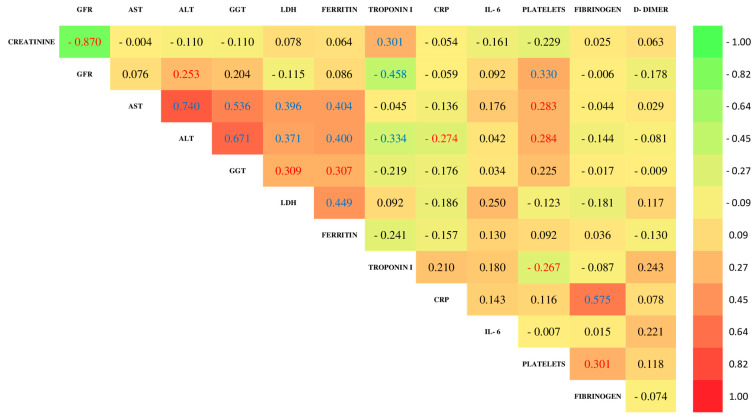
Spearman’s rank correlation coefficients for clinical variables in COVID-19 patients (level of significance: *p* < 0.01 are marked in blue, and *p* < 0.05 are in red. Non-significant values with *p* ≥ 0.05 are marked in black).

**Table 1 jcm-14-03192-t001:** Demographic characteristics of the study population with thrombosis are stratified according to admission by COVID-19 or not.

		*n* (%)/Mean (IC, 95%)	No COVID-19	COVID-19	*p*-Value
Age		66.0 (54.0; 76.5)	66.0 (55.0; 76.2)	63.0 (52.5; 76.0)	0.366 ^a^
Sex	Female	255 (38.5%)	232 (35.0%)	23 (3.5%)	0.738 ^b^
	Male	408 (61.5%)	368 (55.5%)	40 (6.0%)	
Nationality	Spanish	580 (87.5%)	530 (79.9%)	50 (7.5%)	0.054 ^b^
	European (non-Spanish)	47 (7.1%)	42 (6.3%)	5 (0.7%)	
	African	30 (4.5%)	23 (3.5%)	7 (1.0%)	
	South American	6 (0.9%)	5 (0.8%)	1 (0.2%)	
Prior COVID-19	Yes	10 (1.5%)	9 (1.4%)	1 (0.2%)	0.498 ^b^
	Recent (<1 month)	13 (2.0%)	13 (2.0%)	0 (0.0%)	
	No	640 (96.5%)	578 (87.2%)	62 (9.4%)	
Severe COVID-19	Yes	28 (4.2%)	0 (0.0%)	28 (4.2%)	<0.001 ^b^
	No	635 (95.8%)	600 (90.5%)	35 (5.3%)	
ICU	Yes	57 (8.6%)	40 (6.0%)	17 (2.6%)	<0.001 ^b^
	No	606 (91.4%)	560 (84.5%)	46 (6.9%)	

^a^ Mann–Whitney U test. ^b^ Chi-square test.

**Table 2 jcm-14-03192-t002:** Descriptive analysis of the different biomarkers between COVID-19 and non-COVID-19 patients who have suffered a thromboembolic episode.

	No COVID-19							COVID-19							*p*-Value *
	*n*	Mean	SD	CI(95%)	P25	P50	P75	*n*	Mean	SD	CI(95%)	P25	P50	P75	
Creatine	600	0.8	1.2	0.9–1.3	0.6	0.8	1.1	63	0.6	0.6	0.7–1.0	0.5	0.6	0.9	<0.001
GFR	599	87.2	31.1	70.4–84.2	52.3	82.0	98.1	63	100.5	33.8	86.4–105.2	81.7	100.5	118.2	<0.001
AST	572	26.0	83.7	33.4–61.0	20.0	27.5	43.7	63	24.0	36.6	27.5–46.7	19.0	25.0	37.0	0.722
ALT	571	24.0	71.1	31.1–48.4	17.0	24.5	47.7	63	33.0	66.1	37.6–75.9	19.0	33.0	74.0	0.009
GGT	570	56.0	238.9	66.7–124.9	32.0	49.5	116.5	63	55.0	154.8	69.6–164.3	32.0	55.0	119.0	0.596
LDH	394	481.0	435.3	566.5–702.5	464.7	547.5	743.7	58	538.5	257.0	536.2–685.8	409.0	556.0	764.0	0.182
Ferritin	281	300.5	2379.9	377.7–784.4	135.0	307.0	512.5	61	441.0	668.2	497.8–901.3	199.6	415.0	779.6	0.005
Troponin I	232	22.6	1.8	41.2–220.1	10.4	21.9	57.3	59	6.9	119.3	10.7–70.2	3.7	7.6	25.0	<0.001
CRP	596	3.6	8.2	4.7–9.9	1.7	3.4	8.4	63	3.3	7.1	4.9–9.1	1.1	3.6	12.7	0.645
IL-6	94	38.8	363.9	46.4–221.1	12.3	39.2	122.0	56	25.4	701.4	64.3–376.0	9.6	30.3	71.8	0.401
Platelets (×10^5^)	600	2.3	1.3	2.4–2.9	2.0	2.4	3.2	63	2.5	1.5	2.6–3.4	1.9	2.5	4.1	0.205
Fibrinogen	597	626.0	207.7	604.0–691.3	509.7	629.0	765.7	63	614.0	187.4	567.0–683.2	509.0	614.0	747.0	0.520
D-dimer (×10^3^)	379	2.4	12.0	4.9–8.9	1.8	3.8	7.4	60	2.3	15.0	2.9–11.9	1.3	2.2	6.5	0.810

Units: mg/dL for creatinine; ml/min/1.73 m^2^ for GFR; IU/L for GGT, LDH, ALT, and AST; ng/mL for ferritin, troponin I and IL-6; mg/dL for CRP; cell/µL for platelets; and ng/mL for D-dimer. * Mann-Whitney U-test.

**Table 3 jcm-14-03192-t003:** Descriptive analysis of biomarker levels in arterial thromboembolism stratifying by COVID-19 history.

	COVID-19			Non-COVID-19			*p*-Value *
	*n*	Mean	SD	*n*	Mean	SD	
Creatinine	55	0.8	0.6	447	1.0	0.9	0.010
GFR	55	93.8	30.5	447	78.3	30.0	0.010
AST	55	35.0	33.4	447	46.7	64.5	0.915
ALT	55	48.3	52.0	447	39.6	40.6	0.005
GGT	55	116.5	176.5	447	83.8	101.2	0.216
LDH	55	611.7	264.5	447	620.0	280.4	0.152
Ferritin	55	705.0	747.2	447	534.9	901.6	0.004
Troponin I	55	43.3	111.7	447	141.2	419.5	<0.001
CRP	55	6.6	7.5	447	5.9	7.2	0.444
IL-6	55	201.0	557.1	447	140.3	410.4	0.126
Platelets (×10^5^)	55	2.9	1.4	447	2.7	1.1	0.768
Fibrinogen	55	605.7	186.6	447	635.6	198.8	0.261
D-Dimer (×10^3^)	55	7.9	17.0	447	7.2	9.3	0.368

Units: mg/dL for creatinine; ml/min/1.73 m^2^ for GFR; IU/L for GGT, LDH, ALT, and AST; ng/mL for ferritin, troponin I and IL-6; mg/dL for CRP; cell/µL for platelets; and ng/mL for D-dimer. * Mann–Whitney U-test.

**Table 4 jcm-14-03192-t004:** Descriptive analysis of biomarker levels in venous thromboembolism stratifying by COVID-19 history.

	COVID-19			Non-COVID-19			*p*-Value *
	*n*	Mean	SD	*n*	Mean	SD	
Creatinine	14	1.0	0.8	249	1.5	1.5	0.056
GFR	14	93.5	52.4	249	70.4	35.3	0.097
AST	14	50.2	48.6	249	49.2	45.8	0.781
ALT	14	95.2	116.1	249	40.2	34.3	0.480
GGT	14	109.6	89.5	249	112.1	161.5	0.610
LDH	14	589.0	234.9	249	616.5	281.2	0.511
Ferritin	14	573.2	383.1	249	527.3	606.9	0.092
Troponin I	14	35.6	51.9	249	210.5	591.7	0.143
CRP	14	6.7	6.4	249	11.4	17.2	0.965
IL-6	14	249.9	435.8	249	245.8	637.9	0.721
Platelets (×10^5^)	14	3.3	1.2	249	2.5	1.1	0.021
Fibrinogen	14	680.7	239.	249	713.3	197.4	0.720
D-Dimer (×10^3^)	14	6.5	7.1	249	5.8	8.9	0.507

Units: mg/dL for creatinine; ml/min/1.73 m^2^ for GFR; IU/L for GGT, LDH, ALT, and AST; ng/mL for ferritin, troponin I and IL-6; mg/dL for CRP; cell/µL for platelets; and ng/mL for D-dimer. * Mann–Whitney U-test.

**Table 5 jcm-14-03192-t005:** Stepwise multiple binary logistic regression analysis of the risk of ICU admission adjusted for several potential risk factors, including biological markers.

No COVID-19 Patients				COVID-19 Patients			
Parameter	OR	95% C.I.	*p*-Value	Parameter	OR	95% C.I.	*p*-Value
Ferritin	−0.222	0.000–2.02 × 10^13^	0.989	Sex (1)	2.131	1.343–52.872	0.023
Platelets	0.000	0.947–1.057	0.993	Creatine	−3.839	0.000–1.193	0.061

The regression model was adjusted for sex (0: female; 1: male), age, prior COVID-19 (0: No; 1: Yes), creatine, GFR, AST, ALT, GGT, LDH, ferritin, troponin I, PCR, IL-6, platelets, fibrinogen, and D-dimer).

## Data Availability

All data supporting the findings of this study are available from the corresponding author upon reasonable request.

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
