# Peer review of "Hematological Markers in Thromboembolic Events: A Comparative Study of COVID-19 and Non-COVID-19 Hospitalized Patients"

_jcm, 2025, doi:10.3390/jcm14093192_

Round 1

Reviewer 1 Report

Comments and Suggestions for Authors

  1. While the manuscript addresses an important clinical question, how COVID-19 influences biomarkers in patients with thromboembolic events, its novelty could be strengthened by engaging more directly with recent studies that explore platelet heterogeneity in immune-mediated thrombosis. For instance, the study by Qiu et al. (2024) on platelet subpopulations in COVID-19, sepsis offers mechanistic insight into the inflammatory and pro-thrombotic phenotypes observed in severe disease.

Qiu, X., Nair, M.G., Jaroszewski, L. and Godzik, A., 2024. Deciphering Abnormal Platelet Subpopulations in COVID-19, Sepsis and Systemic Lupus Erythematosus through Machine Learning and Single-Cell Transcriptomics. International Journal of Molecular Sciences25(11), p.5941.

  1. The manuscript needs graphical summaries of the data. In the complex correlation patterns presented in Supplementary Table S2 are difficult to interpret in tabular form. I recommend incorporating a correlation heatmap (e.g., using the corrplot package in R: https://cran.r-project.org/web/packages/corrplot/vignettes/corrplot-intro.html) into the main manuscript to make the relationships between biomarkers more readable for readers.

  1. Typos:

- Supplementary Tables S1 and S2 “PATELETS” should be corrected to “PLATELETS” in both column and row headers.

- Supplementary Tables S3–S5 show inconsistent formatting of confidence intervals (CIs). For example: Table entries such as “-1.06–75.56”;  In some rows, CIs are reported as a single number which is nonstandard and unclear.

- Supplementary Table S5.docx Numeric formatting errors like “3007,66.67” and “2944,42.42.

  1. There are multiple discrepancies between the main text and the supplementary tables:

- The text line 160 claims significant positive correlations between "AST and troponin I, CRP, IL-6 and D-dimer" in the COVID-19 group, but Table S2 shows AST is not significantly correlated with troponin I (-0.045, not significant), CRP (-0.136, not significant), or IL-6 (0.176, not significant).

- The text line 161 states a positive correlation between "ALT and troponin I" in COVID-19 patients, but Table S2 shows a significant negative correlation (-0.334**) between them.

- The text claims "GGT and LDH, CRP, and D-dimer" correlations in COVID-19 patients, but Table S2 shows GGT is not significantly correlated with CRP (-0.176) or D-dimer (-0.009).

- line 170 whereas in COVID-19 patients, men had higher D-dimer, but Table S5 actually shows women had much higher D-dimer levels

- line 255-256 claims "COVID-19 patients without ICU admission and male patients exhibited higher D-dimer levels," but Table S5 actually shows females had significantly higher D-dimer levels in COVID-19 patients (13,105.83 vs 4,272.85, p=0.028), contradicting this statement.

  1. The logistic regression model included in the discussion section raises concerns about robustness:

- The extremely wide confidence interval for ferritin in non-COVID-19 patients (0.000–2.02×10¹³) suggests potential data sparsity or multicollinearity issues that may compromise model reliability.

- Similarly, the confidence interval for sex in COVID-19 patients (1.343–52.872), although statistically significant, is very broad and indicates imprecision.

- Please investigate the possibility of overfitting, perform multicollinearity diagnostics.

  1. There are numerous inconsistencies between the narrative descriptions and the supplementary tables. For a scientifically rigorous and transparent manuscript, all textual claims should be cross-checked against tabular data. Any discrepancies must be reconciled, and if necessary, the statistical analyses should be rerun and verified.

Author Response

Thank you for dedicating your valuable time and expertise to reviewing our manuscript. We sincerely appreciate the thoroughness of your feedback and the insightful comments you provided. We have carefully considered each of your suggestions and have made revisions accordingly. We believe that your detailed critique has been instrumental in identifying areas for improvement. As a result, we are confident that your valuable assistance has significantly strengthened the manuscript. We are grateful for your contribution to enhancing the quality of our work.

  1. While the manuscript addresses an important clinical question, how COVID-19 influences biomarkers in patients with thromboembolic events, its novelty could be strengthened by engaging more directly with recent studies that explore platelet heterogeneity in immune-mediated thrombosis. For instance, the study by Qiu et al. (2024) on platelet subpopulations in COVID-19, sepsis offers mechanistic insight into the inflammatory and pro-thrombotic phenotypes observed in severe disease. 

Qiu, X., Nair, M.G., Jaroszewski, L. and Godzik, A., 2024. Deciphering Abnormal Platelet Subpopulations in COVID-19, Sepsis and Systemic Lupus Erythematosus through Machine Learning and Single-Cell Transcriptomics. International Journal of Molecular Sciences, 25(11), p.5941.

REPLY: Thank you very much for this valuable suggestion. The authors have carefully read the article by Qiu et al. (2024) and consider it to be of great interest. Consequently, we have included a discussion on platelet heterogeneity and its relevance in the context of our findings in our manuscript's Discussion section. We believe this addition significantly strengthens the novelty and scope of our work by directly connecting it with recent research in the field.

  1. The manuscript needs graphical summaries of the data. In the complex correlation patterns presented in Supplementary Table S2 are difficult to interpret in tabular form. I recommend incorporating a correlation heat map (e.g., using the corrplot package in R: https://cran.r-project.org/web/packages/corrplot/vignettes/corrplot-intro.html) into the main manuscript to make the relationships between biomarkers more readable for readers.

REPLY: Thank you for highlighting the need for graphical summaries of the data. We agree that the complex correlation patterns in Supplementary Table S1 and Supplementary Table S2 are difficult to interpret in tabular form. Following your valuable suggestion, we have incorporated correlation heat maps into both Supplementary Tables to provide a more intuitive and readable visualization of the relationship between biomarkers. At the same time, they have been transformed into Figures 1 and 2, including them in the text for better visualization, consequently modifying the numbering of the remaining supplementary tables. We believe this modification significantly enhances the clarity and interpretability of our findings.

  1. Typos:

- Supplementary Tables S1 and S2 "PATELETS" should be corrected to "PLATELETS" in both column and row headers.

REPLY: Thank you for pointing out this typo. Following your feedback, we have corrected it in the revised tables (now Figures 1 and 2).

- Supplementary Tables S3–S5 show inconsistent formatting of confidence intervals (CIs). For example: Table entries such as "-1.06–75.56"; In some rows, CIs are reported as a single number which is nonstandard and unclear.

REPLY: Thank you for your thorough review and insightful feedback concerning the formatting of the supplementary tables. We have meticulously examined all tables, including Supplementary Tables S3–S5 (now S1-S3), and made specific enhancements to improve consistency and clarity. Additionally, we have addressed any identified issues throughout the manuscript. Specifically, the inconsistent values (negative and single values, without interval) have been reviewed and corrected. We trust that these modifications will enhance the presentation and understanding of our results.

- Supplementary Table S5.docx Numeric formatting errors like “3007,66.67” and “2944,42.42.

REPLY: We appreciate you pointing out this typo, which has now been corrected in the revised supplementary tables.

  1. There are multiple discrepancies between the main text and the supplementary tables:

- The text line 160 claims significant positive correlations between "AST and troponin I, CRP, IL-6 and D-dimer" in the COVID-19 group, but Table S2 shows AST is not significantly correlated with troponin I (-0.045, not significant), CRP (-0.136, not significant), or IL-6 (0.176, not significant). 

REPLY: We sincerely appreciate your observations regarding the discrepancies between the main text and the tables. We have thoroughly revised the correlation analysis results in light of your comments. To facilitate a better understanding of the correlations, we have transformed the tables into figures that now include visually more intuitive heat maps. This graphical representation allows for a more precise and concise identification of the relationships between the variables.

Furthermore, we have completely rewritten the corresponding results paragraph in the manuscript. We have corrected the significant errors in the writing that you kindly pointed out. We hope the new version of the paragraph clarifies the points you identified and improves the quality and accuracy of the manuscript.

- The text line 161 states a positive correlation between "ALT and troponin I" in COVID-19 patients, but Table S2 shows a significant negative correlation (-0.334**) between them.

REPLY: The same applies as in the previous point.

- The text claims "GGT and LDH, CRP, and D-dimer" correlations in COVID-19 patients, but Table S2 shows GGT is not significantly correlated with CRP (-0.176) or D-dimer (-0.009).

REPLY: The same applies as in the previous point.

- line 170 whereas in COVID-19 patients, men had higher D-dimer, but Table S5 actually shows women had much higher D-dimer levels

REPLY: We greatly appreciate your identification of the typo on line 170. This was indeed a typographical error in the main text, and we have promptly taken action to review and correct the statement. The revised text accurately reflects the data in Table S5 (now Table S3), which demonstrates significantly higher D-dimer levels in women in the COVID-19 group.

- line 255-256 claims "COVID-19 patients without ICU admission and male patients exhibited higher D-dimer levels," but Table S5 actually shows females had significantly higher D-dimer levels in COVID-19 patients (13,105.83 vs 4,272.85, p=0.028), contradicting this statement.

REPLY: Thank you for pointing out the inconsistency. You are correct in identifying the contradiction between the statement in lines 255-256 and the data presented in Table S5 (now updated to Table S3). This was a typographical error, and we have promptly made the necessary corrections in the manuscript.

  1. The logistic regression model included in the discussion section raises concerns about robustness:

- The extremely wide confidence interval for ferritin in non-COVID-19 patients (0.000–2.02×10¹³) suggests potential data sparsity or multicollinearity issues that may compromise model reliability.

- Similarly, the confidence interval for sex in COVID-19 patients (1.343–52.872), although statistically significant, is very broad and indicates imprecision.

- Please investigate the possibility of overfitting, perform multicollinearity diagnostics.

REPLY: We appreciate the suggestion to investigate potential overfitting and conduct multicollinearity diagnostics. In our manuscript's Methodology and Results sections, we calculated the Variance Inflation Factors (VIFs) for each independent variable included in our model to assess multicollinearity. Our analysis revealed maximum VIF values around 6.   This information and the specific steps taken to address any potential multicollinearity have been thoroughly documented in the respective sections of the manuscript, making it easy for you to verify our process. Additionally, we have acknowledged the imbalance in sample size between the non-COVID-19 and COVID-19 groups by including it as a limitation at the end of the Discussion. This proactive approach reflects our confidence in the foresight of our research, as we identified this difference as a potential source of overfitting in our model.

  1. There are numerous inconsistencies between the narrative descriptions and the supplementary tables. For a scientifically rigorous and transparent manuscript, all textual claims should be cross-checked against tabular data. Any discrepancies must be reconciled, and if necessary, the statistical analyses should be rerun and verified.

REPLY: Thank you for your important observation regarding the inconsistencies between the narrative descriptions and the supplementary tables. We have taken your recommendation concerning cross-verification very seriously to ensure the rigor and transparency of our manuscript.

Based on your feedback, we have comprehensively and meticulously reviewed all textual claims regarding the data presented in the supplementary tables. We have reconciled all identified discrepancies, correcting the text and the tables as necessary to ensure consistency.

When inconsistencies required re-evaluating the statistical analyses, we re-ran them and verified the results to ensure their accuracy. The revised supplementary tables now reflect these verified analyses. These corrections have significantly strengthened the integrity and clarity of our manuscript.

Reviewer 2 Report

Comments and Suggestions for Authors

This retrospective observational study compares key hematological, hepatic, renal, inflammatory, and coagulation markers between hospitalized patients with thromboembolic events, with or without COVID-19 infection. It provides a valuable dataset drawn from a substantial patient population (n=663), with insightful subgroup analyses that include ICU admission, biomarker correlations, and sex- and age-based differences. However, several issues regarding interpretation, clarity, and scientific positioning should be addressed to improve the quality of the manuscript.

1. Interpretation of GFR differences lacks mechanistic clarity

The finding of higher GFR in COVID-19 patients, particularly those admitted to ICU, appears counterintuitive, given that acute kidney injury is a known complication of severe COVID-19. Although the authors speculate that comorbidities and age might confound this result, this section lacks mechanistic depth. For example, the potential influence of fluid overload, glomerular hyperfiltration, or altered creatinine kinetics in COVID-19 patients should be explored. Including eGFR distribution curves or creatinine clearance data (if available) could strengthen the renal function analysis.

2. Statistical power imbalance due to group size disparity

The study features a significant imbalance in sample size between the non-COVID group (n=600) and COVID-19 group (n=63). This difference reduces the robustness of between-group comparisons, especially in multivariate models and biomarker distribution analyses. Some non-significant findings (e.g., IL-6, CRP, D-dimer) may reflect insufficient power rather than true similarity. A sensitivity analysis or subgroup matching (e.g., age, sex, ICU admission) could mitigate these concerns.

3. Ambiguity in the association between troponinI and COVID status

The observation that troponin I levels are higher in non-COVID-19 patients contradicts previous literature suggesting COVID-19 is a strong driver of myocardial injury. While this is an intriguing finding, the discussion does not convincingly explore potential explanations—such as pre-existing cardiac conditions, differential use of anticoagulants, or timing of biomarker measurement. It is essential to elaborate whether elevated troponin is a consequence of ischemia, systemic inflammation, or cardiac stress unrelated to SARS-CoV-2.

4. No distinction between arterial and venous thrombotic events

The manuscript refers generally to “thromboembolic events” but does not consistently differentiate between arterial (e.g., stroke, MI) and venous (e.g., DVT, PE) events in biomarker analysis. Given the different pathophysiologies and prognostic implications, subgrouping patients by thrombotic event type could reveal more meaningful associations (e.g., D-dimer's role in VTE vs. platelet activation in arterial thrombosis).

Minor Issues and Recommendations:

1. Clinical Relevance: More emphasis could be placed on how findings may influence thromboprophylaxis strategies or biomarker monitoring protocols in clinical settings.

2. Supplementary Materials: The supplementary tables are rich in detail but hard to interpret without referencing the main text. Please provide brief context or highlight key findings in the main discussion.

Author Response

This retrospective observational study compares key hematological, hepatic, renal, inflammatory, and coagulation markers between hospitalized patients with thromboembolic events, with or without COVID-19 infection. It provides a valuable dataset drawn from a substantial patient population (n=663), with insightful subgroup analyses that include ICU admission, biomarker correlations, and sex- and age-based differences. However, several issues regarding interpretation, clarity, and scientific positioning should be addressed to improve the quality of the manuscript.

REPLY: Thank you for your time and dedication in reviewing our manuscript. We greatly appreciate your constructive feedback and acknowledge the importance of the issues you have raised regarding interpretation, clarity, and the scientific positioning of our work. We are pleased that you found the dataset and the subgroup analyses valuable. We take your observations very seriously and are committed to addressing each of them thoroughly to enhance the quality of our manuscript.

  1. Interpretation of GFR differences lacks mechanistic clarity

The finding of higher GFR in COVID-19 patients, particularly those admitted to ICU, appears counterintuitive, given that acute kidney injury is a known complication of severe COVID-19. Although the authors speculate that comorbidities and age might confound this result, this section lacks mechanistic depth. For example, the potential influence of fluid overload, glomerular hyperfiltration, or altered creatinine kinetics in COVID-19 patients should be explored. Including eGFR distribution curves or creatinine clearance data (if available) could strengthen the renal function analysis.

REPLY: We are most grateful for the insightful observation regarding the need for enhanced mechanistic clarity in our interpretation of the observed differences in GFR. We sincerely appreciate your feedback and have carefully considered this point. In response, we have undertaken a thorough review of the existing literature. Consequently, we have integrated an additional paragraph into our manuscript's Discussion section, which includes five additional references. This expanded section delves into the potential underlying mechanisms that could contribute to our findings, explicitly addressing the possible influence of factors such as fluid overload and glomerular hyperfiltration within the unique pathophysiological context of COVID-19.

Additionally, we acknowledge the significant value that data on GFR distribution curves and detailed creatinine clearance would bring to a more granular analysis of renal function in our cohort. Regrettably, this specific information was not systematically available due to the exigencies of the clinical environment during the pandemic and the retrospective nature of our data collection process. Despite this constraint, we are confident that including a more detailed mechanistic discussion significantly enhances the rigor and interpretability of our findings, providing a more nuanced understanding of the observed renal function patterns. This confidence in our enhanced discussion should reassure you about the quality of our research.

  1. Statistical power imbalance due to group size disparity

The study features a significant imbalance in sample size between the non-COVID group (n=600) and COVID-19 group (n=63). This difference reduces the robustness of between-group comparisons, especially in multivariate models and biomarker distribution analyses. Some non-significant findings (e.g., IL-6, CRP, D-dimer) may reflect insufficient power rather than true similarity. A sensitivity analysis or subgroup matching (e.g., age, sex, ICU admission) could mitigate these concerns.

REPLY: We sincerely appreciate your feedback regarding the imbalance in the sizes of the study groups. We acknowledge the disparity between the non-COVID-19 group (n=600) and the COVID-19 group (n=63), and we understand your concerns about statistical power. As we emphasize in the Discussion section, this significant difference in sample size arose from the methodological and contextual challenges posed by the COVID-19 pandemic during the recruitment phase. Public health restrictions and safety concerns greatly impacted the availability and participation of individuals in the COVID-19 group, a challenge recognized in the literature (Butler et al., 2023). This limitation may have affected the statistical power for detecting significant differences in specific analyzed biomarkers. Nevertheless, the findings of our study are valuable, and we have thoroughly addressed this issue in the Discussion section to provide appropriate context for interpreting our results.

Butler, A. M., Burcu, M., Christian, J. B., Tian, F., Andersen, K. M., Blumentals, W. A., Joynt Maddox, K. E., & Alexander, G. C. (2023). Noninterventional studies in the COVID-19 era: methodological considerations for study design and analysis. Journal of clinical epidemiology, 153, 91–101. https://doi.org/10.1016/j.jclinepi.2022.11.011

  1. Ambiguity in the association between troponin I and COVID status

The observation that troponin I levels are higher in non-COVID-19 patients contradicts previous literature suggesting COVID-19 is a strong driver of myocardial injury. While this is an intriguing finding, the discussion does not convincingly explore potential explanations—such as pre-existing cardiac conditions, differential use of anticoagulants, or timing of biomarker measurement. It is essential to elaborate whether elevated troponin is a consequence of ischemia, systemic inflammation, or cardiac stress unrelated to SARS-CoV-2.

REPLY: We sincerely appreciate your comment regarding the elevated troponin I levels in the non-COVID-19 group. We acknowledge that this differs from previous literature emphasizing COVID-19 as a significant driver of myocardial injury. In response to your concern, we have included the following paragraph in the discussion:

"Unlike other studies, this study found elevated troponin I levels in non-COVID-19 patients, which could be attributed to the higher prevalence of comorbidities and cardiovascular risk factors in this group. Chronic ischemic heart disease, smoking, anticoagulant use, and liver disease could induce or exacerbate troponin release. Non-cardiac surgical interventions, prolonged immobilization, and sepsis, more frequent in non-COVID-19 patients, also contribute to myocardial damage and troponin elevation. In contrast, although COVID-19 can elevate troponin, associated comorbidities are less common in this group. Therefore, the elevated troponin I in non-COVID-19 patients in this cohort appears to be more related to their cardiovascular risk profile and comorbidities than to the viral infection itself."

We hope this elaboration clarifies the potential explanations behind our findings and addresses your concern about the apparent contradiction with previous literature.

  1. No distinction between arterial and venous thrombotic events

The manuscript refers generally to "thromboembolic events" but does not consistently differentiate between arterial (e.g., stroke, MI) and venous (e.g., DVT, PE) events in biomarker analysis. Given the different pathophysiologies and prognostic implications, subgrouping patients by thrombotic event type could reveal more meaningful associations (e.g., D-dimer's role in VTE vs. platelet activation in arterial thrombosis).

REPLY: We sincerely appreciate your insightful observation about the need to differentiate between arterial and venous thrombotic events in our biomarker analysis. These events' distinct pathophysiologies and prognostic implications make subgroup analysis essential.

Following your valuable suggestion, we conducted an additional analysis that specifically separates patients who experienced arterial from those who experienced venous thromboembolic events. This enhancement, as you recommended, has significantly improved the quality and depth of our manuscript.

In this sense, two new tables have been incorporated to present the obtained results, which are described in detail therein. Furthermore, the following paragraph has been added to the discussion section:

"Analysis of thrombotic territories in COVID-19 patients revealed notable differences in various biomarkers. In cases of arterial thrombosis, alterations in creatinine, GFR, ALT, ferritin, and troponin were identified, likely attributable to systemic inflammation, endothelial dysfunction, and coagulopathy induced by COVID-19. Conversely, venous thrombosis exhibited prominent elevations in creatinine and platelet counts, which may correlate with platelet hyperreactivity and acute kidney injury linked to SARS-CoV-2 infection. In contrast, thrombosis among non-COVID-19 patients tended to follow a more traditional multifactorial profile. Interestingly, D-dimer levels did not show significant differences between the two groups in either territory. This observation may stem from distinct underlying mechanisms and the effects of comorbidities and treatments in the non-COVID-19 cohort. These findings underscore that thrombosis associated with COVID-19 represents a unique clinical phenotype driven by specific immunothrombotic mechanisms."

As you rightly pointed out, we are confident that this new stratification will enhance our understanding of the connections between the biomarkers and the types of thrombotic events.

Minor Issues and Recommendations:

  1. Clinical Relevance: More emphasis could be placed on how findings may influence thromboprophylaxis strategies or biomarker monitoring protocols in clinical settings.

REPLY: We sincerely appreciate your suggestion. We have noted this observation and made the appropriate modifications to improve the clarity and impact of our work.

Regarding your recommendation on the clinical relevance of our findings, we have introduced the following sentence in the Conclusions section to address this point more explicitly:

"Integrating individualized risk assessment models and biomarker monitoring into thromboprophylaxis protocols can improve patient outcomes by tailoring prophylactic strategies to the specific risk profiles of hospitalized patients."

We hope this addition clarifies the potential clinical utility of our findings and strengthens the discussion of their relevance.

  1. Supplementary Materials: The supplementary tables are rich in detail but hard to interpret without referencing the main text. Please provide brief context or highlight key findings in the main discussion.

REPLY: We appreciate your insightful feedback regarding the presentation of our supplementary tables. Recognizing the challenges associated with interpreting these materials in isolation from the main text, we have undertaken several modifications to enhance their accessibility and clarity.

Responding to your concerns, we have meticulously reviewed all supplementary tables to streamline their presentation and ensure conciseness. Furthermore, we have incorporated a brief contextual summary within the manuscript to underscore the key findings derived from these tables, facilitating a more straightforward interpretation.

In addition, we have included a heatmap to illustrate the correlation analyses among biomarkers. This visual representation enables rapid identification of patterns and correlation intensities, thus providing a comprehensive overview of the data that complements the detailed information in the tables.

We trust that these enhancements will significantly improve the interpretability of the supplementary materials and contribute to a deeper understanding of our findings.

Round 2

Reviewer 1 Report

Comments and Suggestions for Authors

The authors revised accordingly. 

Author Response

Dear Reviewer,

We want to express our sincere gratitude for the time and effort you dedicated to reviewing our manuscript.

We have carefully considered all of your insightful comments and suggestions. The revisions based on your feedback have improved the quality and clarity of our manuscript. Your expertise and constructive criticism have been invaluable in refining our work.

Thank you once again for your valuable contribution to the peer-review process.